# Different structural variant prediction tools yield considerably different results in *Caenorhabditis elegans*

Kyle Lesack[1,2], Grace M. Mariene[1,2], Erik C. Andersen[3], James D. Wasmuth[1,2]*

**1** Faculty of Veterinary Medicine, University of Calgary, Alberta, Canada, **2** Host-Parasite Interactions Research Training Network, University of Calgary, Alberta, Canada, **3** Department of Molecular Biosciences, Northwestern University, Evanston, IL, United States of America

* jwasmuth@ucalgary.ca

**Data Availability Statement:** Custom Python (v3.7) and Bash scripts are available at https://github.com/kyleLesack/sv_calling_benchmarking. The repository includes the scripts used to analyze

## Abstract

The accurate characterization of structural variation is crucial for our understanding of how large chromosomal alterations affect phenotypic differences and contribute to genome evolution. Whole-genome sequencing is a popular approach for identifying structural variants, but the accuracy of popular tools remains unclear due to the limitations of existing benchmarks. Moreover, the performance of these tools for predicting variants in non-human genomes is less certain, as most tools were developed and benchmarked using data from the human genome. To evaluate the use of long-read data for the validation of short-read structural variant calls, the agreement between predictions from a short-read ensemble learning method and long-read tools were compared using real and simulated data from *Caenorhabditis elegans*. The results obtained from simulated data indicate that the best performing tool is contingent on the type and size of the variant, as well as the sequencing depth of coverage. These results also highlight the need for reference datasets generated from real data that can be used as 'ground truth' in benchmarks.

## Introduction

Large alterations in chromosome structure contribute substantially to the genetic diversity observed in natural populations and play a fundamental role in the evolution of novel genes [1]. These changes that span large segments of the genome (*e.g.*, > 100 bp) are termed structural variants (SVs) and include deletions, tandem and interspersed duplications, insertions, and inversions. SVs may be neutral, deleterious, or adaptive [2], and are known to facilitate speciation [3]. SVs drive genome evolution using several mechanisms. For example, large heterozygous inversions can suppress recombination, thereby protecting locally adapted alleles [4]. Also, copy number variation (CNV) is an important factor in genome evolution that describes the gain or loss of genes. CNVs are associated with a wide range of phenotypic effects due to the modulation of gene expression, including differential drug responses between individuals [5], HIV susceptibility [6], autism spectrum disorders [7], and schizophrenia [8]. Gene duplication and subsequent diversification is a source of novel genes and functional

both the short- and long-read data. The mock genome data constitutes terabytes of data and is impractical to store. Therefore, we have included a tutorial with the steps required to recreate the mock genomes.

**Funding:** JDW; 04589-2020; The Natural Sciences and Engineering Research Council of Canada; https://www.nserc-crsng.gc.ca JDW; 06239-2015; The Natural Sciences and Engineering Research Council of Canada; https://www.nserc-crsng.gc.ca JDW; 413888-2012; The Natural Sciences and Engineering Research Council of Canada; https://www.nserc-crsng.gc.ca JDW; 2016F013R; Results Driven Agricultural Research (Alberta); https://rdar.ca/ ECA; no number; National Science Foundation (USA); https://www.nsf.gov/ The funders had no role in study design, data collection and analysis, decision to publish, or preparation of the manuscript.

**Competing interests:** The authors have declared that no competing interests exist.

diversification [9–11]. Importantly, SVs have also been proposed as a source of "missing heritability" seen in genome-wide association studies [12].

Several approaches have been developed for the detection of SVs from paired-end whole-genome sequencing (WGS) data [13, 14]. Paired-end approaches detect SVs when the orientation of a mapped read-pair is inconsistent with the reference genome or when the alignment produces an unexpected insert size. Split-read approaches detect SVs by identifying individual reads that span a given variant, resulting in at least two partial alignments. CNVs may be detected using read-depth differences caused by gene loss or gain. Hybrid approaches that combine multiple signals are employed by many tools and, recently, ensemble methods that leverage multiple separate SV callers have been developed [15, 16].

The emergence of international 'sequence everything' projects [17, 18] and continual reduction in sequencing costs will enable researchers to study the role that SV plays in the evolution of their favourite species. Although numerous tools are available, most benchmarks are limited to the human genome and a limited range of sequencing depths [13, 14, 19, 20]. Due to the difficulty in validating SVs, benchmarks often rely upon simulated data or incomplete sets of experimentally validated variants [21]. Long-read support has also been used to validate SV calls [22], but the accuracy of this method is unknown. Because genome properties, such as repeat content, GC content, or heterozygosity, can differ significantly between species, it is unclear how callers benchmarked on the human genome will perform for other species. At the time of writing, most curated, whole genome, multi-population/isolate sequence datasets are restricted to humans or microbial species. Among metazoans, a notable exception is the *Caenorhabditis elegans* Natural Diversity Resource (CeNDR) which houses whole genome data for 1514 isolates and 548 strains of the free living nematode, *C. elegans* (August 2020) [23]. Because the *C. elegans* genome differs from the human genome in several key characteristics, evaluating SV calling in *C. elegans* would provide valuable points of comparison with past benchmarks. For example, although the GC content in *C. elegans* (36%) [24] does not differ considerably from humans (41%) [25], it is distributed uniformly among the chromosomes, whereas the GC content in the human genome varies considerably depending on the location. Furthermore, *C. elegans* reproduces primarily through self-fertilizing hermaphrodites, which can lead to profound genomic differences compared to dioecious species that reproduce solely through the mating of males and females. Notably, selfing results in increased homozygosity and reduced recombination, which have been predicted to impact transposon dynamics and the purging of deleterious alleles [26]. In terms of repeat content, the *C. elegans* genome contains substantially fewer transposons (~12% of the genome) [27] and other repetitive sequences compared to the repeat-rich human genome (~44% of the genome is derived from transposable elements) [28]. This is a key factor, as repeats are associated with structural variation [29, 30] but are also a source of error in SV calling [14, 19, 31].

Here, the performance of SV calling in *Caenorhabditis elegans* was evaluated using real and simulated data. Multiple commonly used short- and long-read SV calling tools were benchmarked using mock genomes containing simulated variants. Real data were used to assess the degree of overlap between individual callers and to determine if the results from an ensemble short-read approach were comparable to those obtained by long-read callers. The benchmarks and comparisons shown here demonstrate that SV prediction depends highly on the tool used and that optimal tool choice for each platform depends on the type and size of SV. These results provide valuable information for researchers studying structural variation in *C. elegans* or other species with similar genome properties.

## Results

### Structural variation prediction using simulated short-read data

To evaluate the performance of various short-read structural variant calling methods, SVsim was used to create mock genomes containing simulated deletions, duplications, and inversions. Two datasets were created from the *C. elegans* reference genome: a training dataset used to train the FusorSV fusion model, and a testing dataset used to benchmark the performance of each caller. Each dataset contained simulated SVs that ranged from 100bp to 280kbp. Because FusorSV uses variable number of bins to discriminate SVs by size and type during the training phase, 129 mock genomes were created for each dataset so that each bin contained a total of 30 SVs, while ensuring that the total number of base pairs spanned by SVs in each mock genome did not exceed 1% of the *C. elegans* reference genome. The mock genomes were used to generate simulated Illumina reads at 5X, 15X, 30X, and 60X sequencing depth of coverage.

The caller performance varied by variant type and depth (Table 1). For deletion calls, DELLY had the highest F-measure scores at all sequencing depths, followed closely by BreakDancer. The sequencing depth had a stronger impact on the other variant callers, except for Hydra. The accuracies for cnMOPS and CNVnator all improved considerably above 5X, while increased false positives accounted for the decreased accuracy in Lumpy at 60X. The performance of FusorSV was similar the best performing tools for each variant type at all depths.

No single metric completely describes the performance of each variant caller. Because the precision, recall, and F1 scores were calculated based on the number of true positives, false positives, and false negatives, they fail to describe the performance of each caller at the base pair level. The Jaccard similarity score was used to describe the base pair overlap between the predicted variants and simulated variants. Because the Jaccard score is based on the proportion of intersecting base pairs, the metric is biased towards the performance of larger SVs. It should be noted that the overall Jaccard scores described here are biased towards the performance for larger variants, especially the overall values calculated using the entire range of SV sizes. The Jaccard metric was also used to quantify the performance of each caller at 60X sequencing depth for a range of SV size ranges. Although the impact of size on the Jaccard score would be decreased for each range compared to overall score calculated from the entire set of predicted SVs, the metric would remain biased towards the performance for the larger SV sizes contained in each range.

Although DELLY had the highest accuracy at all depths, its Jaccard value decreased from 0.99 at 30X depth to 0.74 at 60X depth (Fig 1). This decrease was caused by a 2.56Mbp false positive at the 60X depth. Differences between the Jaccard and F1 scores were observed for several sizes. At 5X depth, the CNVnator F1 and Jaccard scores were 0.53 and 0.91 respectively (S1 Table). Because CNVnator performed well at predicting larger variants, a higher Jaccard score was obtained despite low accuracy for smaller deletion sizes. Conversely, the Hydra F1 scores ranged between 0.72 and 0.79, while its Jaccard scores ranged between 0.12 and 0.13. The large discrepancies between the F1 score and Jaccard scores resulted from good performance for smaller deletions but poor performance for larger sizes.

For duplication calls, DELLY had the highest F-measure scores at all depths, followed closely by BreakDancer. The accuracies of CNVnator, Hydra, and Lumpy improved with increased depth between 5X and 30X, while only a slight decrease was observed for Hydra at 60X depth. Increased depth decreased the cnMOPS accuracy due to higher false positive rates. The performance of FusorSV was similar the best performing tools for each variant type at all depths.

**Table 1. Caller performance using simulated short-read data.**

| Caller | Depth | Deletions | | | Duplications | | | Inversions | | |
|---|---|---|---|---|---|---|---|---|---|---|
| | | Precision | Recall | F1-Score | Precision | Recall | F1-Score | Precision | Recall | F1-Score |
| BreakDancer | 5X | 0.93 | 0.76 | 0.84 | 0.99 | 0.75 | 0.85 | 0.95 | 0.90 | 0.92 |
| cnMOPS[2] | 5X | 0.83 | 0.32 | 0.46 | 0.61 | 0.74 | 0.67 | NA | NA | NA |
| CNVnator[2] | 5X | **1.0** | 0.36 | 0.53 | **1.00** | 0.65 | 0.79 | NA | NA | NA |
| DELLY | 5X | **1.0** | **0.85** | **0.92** | **1.00** | **0.87** | **0.93** | 0.96 | **0.94** | **0.95** |
| Hydra | 5X | 0.93 | 0.66 | 0.78 | 0.36 | 0.26 | 0.30 | 0.98 | 0.28 | 0.43 |
| Lumpy | 5X | **1.0** | 0.76 | 0.87 | **1.00** | 0.75 | 0.86 | **1.00** | 0.90 | 0.94 |
| FusorSV[1] | 5X | 1.0 | 0.84 | 0.92 | 1.00 | 0.88 | 0.94 | 0.98 | 0.92 | 0.95 |
| BreakDancer | 15X | 0.96 | 0.89 | 0.92 | **1.00** | 0.86 | 0.93 | 0.93 | 0.91 | 0.92 |
| cnMOPS[2] | 15X | 0.86 | 0.62 | 0.72 | 0.35 | 0.79 | 0.48 | NA | NA | NA |
| CNVnator[2] | 15X | **1.00** | 0.66 | 0.79 | 0.99 | 0.73 | 0.84 | NA | NA | NA |
| DELLY | 15X | **1.00** | **0.96** | **0.98** | **1.00** | **0.92** | **0.96** | 0.94 | **0.95** | 0.95 |
| Hydra | 15X | 0.89 | 0.71 | 0.79 | 0.56 | 0.28 | 0.37 | 0.40 | 0.02 | 0.04 |
| Lumpy | 15X | **1.00** | 0.93 | 0.96 | **1.00** | 0.87 | 0.93 | **1.00** | 0.93 | **0.97** |
| FusorSV[1] | 15X | 1.00 | 0.94 | 0.97 | 1.00 | 0.94 | 0.97 | 0.99 | 0.93 | 0.96 |
| BreakDancer | 30X | 0.95 | 0.88 | 0.92 | **1.00** | 0.86 | 0.93 | 0.93 | 0.91 | 0.92 |
| cnMOPS[2] | 30X | 0.73 | 0.78 | 0.76 | 0.20 | 0.82 | 0.32 | NA | NA | NA |
| CNVnator[2] | 30X | 0.99 | 0.79 | 0.88 | 0.99 | 0.8 | 0.88 | NA | NA | NA |
| DELLY | 30X | **1.00** | **0.97** | **0.98** | 0.99 | **0.93** | **0.96** | 0.94 | **0.96** | 0.95 |
| Hydra | 30X | 0.85 | 0.71 | 0.77 | 0.77 | 0.27 | 0.40 | 0.27 | 0.02 | 0.04 |
| Lumpy | 30X | 0.99 | 0.93 | 0.96 | **1.00** | 0.90 | 0.95 | **1.00** | 0.94 | **0.97** |
| FusorSV[1] | 30X | 1.00 | 0.94 | 0.97 | 1.00 | 0.93 | 0.97 | 0.98 | 0.92 | 0.96 |
| BreakDancer | 60X | 0.94 | 0.88 | 0.91 | **1.00** | 0.86 | 0.93 | 0.93 | 0.92 | 0.92 |
| cnMOPS[2] | 60X | 0.48 | 0.86 | 0.62 | 0.11 | 0.88 | 0.20 | NA | NA | NA |
| CNVnator[2] | 60X | 0.81 | 0.88 | 0.84 | 0.95 | 0.82 | 0.88 | NA | NA | NA |
| DELLY | 60X | **0.98** | **0.97** | **0.98** | 0.99 | **0.93** | **0.96** | 0.93 | **0.95** | 0.94 |
| Hydra | 60X | 0.79 | 0.67 | 0.72 | 0.77 | 0.26 | 0.38 | 0.21 | 0.01 | 0.03 |
| Lumpy | 60X | 0.41 | 0.94 | 0.57 | **1.00** | 0.91 | 0.95 | **1.00** | 0.94 | **0.97** |
| FusorSV[1] | 60X | 1.00 | 0.95 | 0.97 | 1.00 | 0.93 | 0.96 | 0.99 | 0.94 | 0.97 |

1. FusorSV used a training model trained on simulated data for the other callers.

2. Tool doesn't predict inversions.

The performance of cnMOPS and Hydra were both dependent on the size of the predicted duplications. The rate of false positive duplications increased with higher sequencing depths for cnMOPS and was biased towards smaller duplication sizes (Fig 2). All predicted duplications from Hydra were below 10 kbp, which led to a Jaccard score of 0.01 at all depths (S2 Table). At 60X depth, DELLY predicted a 2.32 Mbp false positive duplication, which resulted in a decreased Jaccard score (0.83).

BreakDancer, DELLY, and Lumpy performed well for the prediction of inversions, as the accuracy of each tool was at least 0.92 at all depths. The accuracy for Hydra was considerably lower at 5X and decreased with increasing depth. The precision, recall, and F1 score for FusorSV was similar to the best performing tools for each variant type at all depths. However, lower Jaccard scores were observed in FusorSV (S3 Table) due to several multiple megabase spanning false positives that were not predicted by other callers.

The performance of BreakDancer, Hydra, and DELLY were dependent on the size of the predicted inversions (Fig 3). Both BreakDancer and DELLY performed better for higher

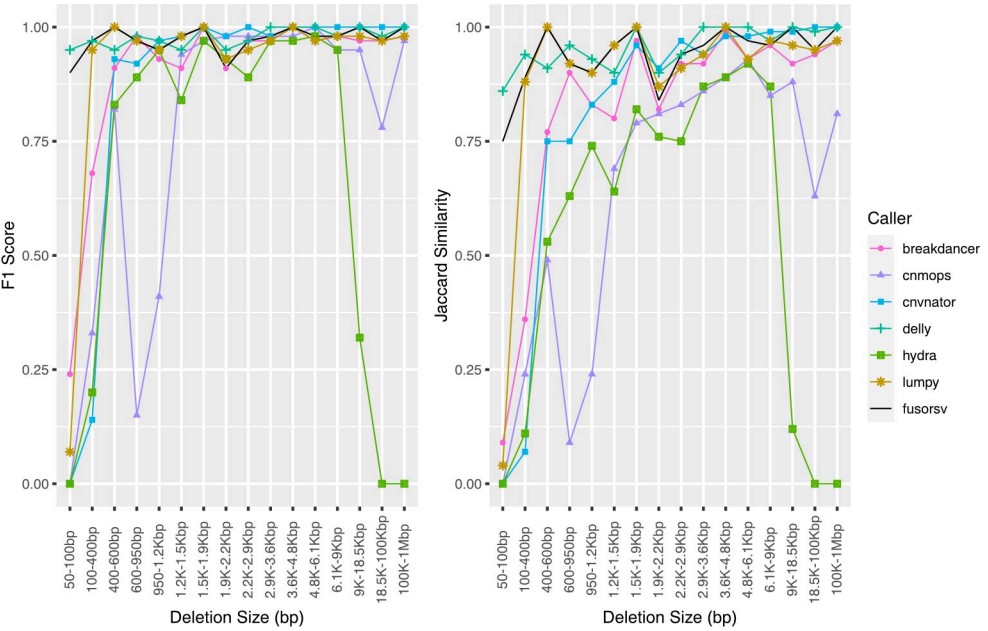

**Fig 1. Accuracy of predicted deletions from simulated short-read data.** Results shown for 60X depth.

inversion sizes, while the performance of Hydra decreased with increasing size. A 4.27 Mbp false positive in DELLY at 15X, 30X, and 60X resulted in decreased Jaccard scores.

## Prediction of known structural variants in C. elegans

BC4586 is a *C. elegans* strain containing experimentally validated structural variants [32]. Publicly available Illumina sequencing data allowed us to determine if the short-read SV callers in

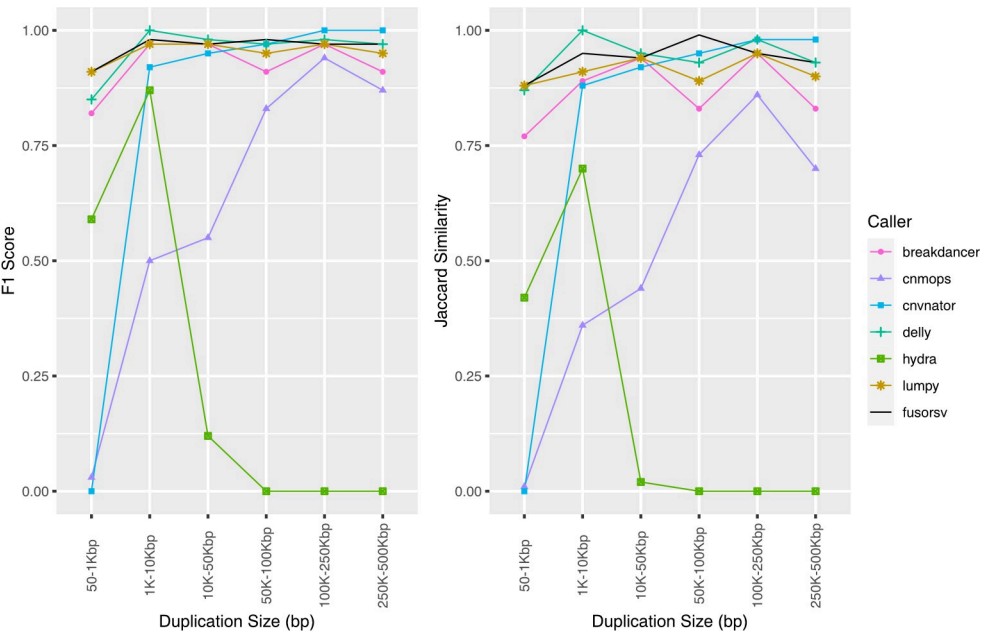

**Fig 2. Accuracy of predicted duplications from simulated short-read data.** Results shown for 60X depth.

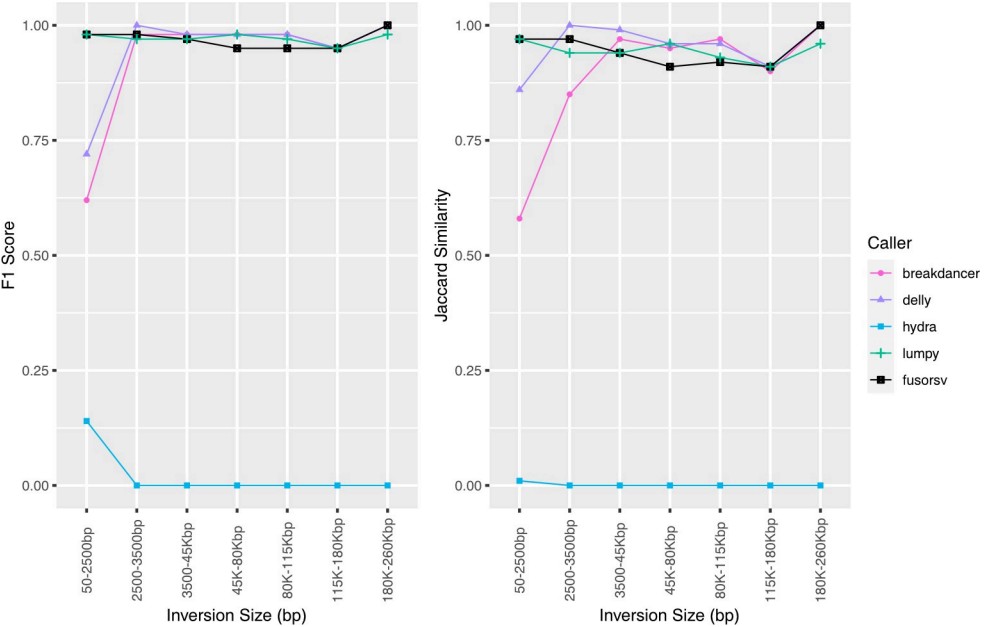

**Fig 3. Accuracy of predicted inversions from simulated short-read data.** Results shown for 60X depth.

the SVE/FusorSV pipeline can resolve a 3910bp deletion (DEL-1; IV:9,853,675–9,857,585), a 552bp tandem duplication (DUP-1; IV:9,853,123–9,853,675), and a 4812bp inversion (INV-1; IV:9,857,585–9,862,397). Only cnMOPs and CNVnator were able to resolve the deletion (Table 2). Among the callers capable of predicting inversions, BreakDancer, DELLY, and Hydra predicted the inversion. Each caller predicted the tandem duplication. Both Break-Dancer and DELLY predicted multiple overlapping inversions spanning INV-1.

## Structural variation prediction using simulated long-read data

Simulated PacBio DNA sequencing data was used to evaluate using long-read sequencing to validate SV calls generated from short-read sequencing platforms. The mock genomes containing simulated deletions, tandem duplications, and inversions were used to generate simulated PacBio reads at 5X, 15X, 30X, 60X, and 142X depth of coverage.

**Table 2. Identification of known variants in *C. elegans*.**

| Caller | DEL-1 | INV-1 | DUP-1 |
|---|---|---|---|
| BreakDancer | No | Yes[1] | Yes |
| cnMOPs | Yes | N/A[2] | Yes |
| CNVnator | Yes | N/A[2] | Yes |
| DELLY | No | Yes[3] | Yes |
| Hydra | No | Yes | Yes |
| Lumpy | No | No | Yes |
| FusorSV | No | Yes | Yes |

1. BreakDancer predicted two inversions spanning the INV-1 genome coordinates
2. Neither cnMOPs nor CNVnator predict inversions.
3. DELLY predicted three inversions spanning the INV-1 genome coordinates

**Table 3. Performance of long-read structural variant callers on simulated data.**

| Caller | Depth | Deletions | | | Duplications | | | Inversions | | |
|---|---|---|---|---|---|---|---|---|---|---|
| | | Precision | Recall | F1-Score | Precision | Recall | F1-Score | Precision | Recall | F1-Score |
| pbsv | 5X | 0.49 | 0.58 | 0.53 | **1.00** | 0.22 | 0.36 | **1.00** | 0.52 | **0.69** |
| Sniffles | 5X | **1.00** | 0.03 | 0.05 | **1.00** | 0.02 | 0.03 | **1.00** | 0.23 | 0.38 |
| SVIM | 5X | 0.86 | **0.84** | **0.85** | 0.79 | **0.32** | **0.45** | 0.46 | **0.54** | 0.49 |
| pbsv | 15X | 0.41 | **0.91** | 0.57 | **1.00** | 0.23 | 0.38 | **1.00** | 0.59 | 0.75 |
| Sniffles | 15X | **0.99** | 0.86 | 0.92 | **1.00** | **0.83** | **0.91** | **1.00** | **0.96** | **0.98** |
| SVIM | 15X | 0.98 | **0.91** | **0.95** | **1.00** | 0.40 | 0.57 | 0.84 | 0.67 | 0.74 |
| pbsv | 30X | 0.44 | 0.92 | 0.60 | **1.00** | 0.30 | 0.46 | **1.00** | 0.59 | 0.75 |
| Sniffles | 30X | **0.99** | **0.99** | **0.99** | **1.00** | **0.97** | **0.98** | 0.89 | **0.96** | **0.92** |
| SVIM | 30X | **0.99** | 0.93 | 0.96 | **1.00** | 0.43 | 0.61 | 0.92 | 0.65 | 0.76 |
| pbsv | 60X | 0.68 | 0.93 | 0.78 | **1.00** | 0.38 | 0.55 | **1.00** | 0.59 | **0.75** |
| Sniffles | 60X | 0.87 | **1.00** | 0.93 | 0.91 | **0.97** | **0.94** | 0.33 | **0.97** | 0.49 |
| SVIM | 60X | 0.98 | 0.93 | **0.96** | **1.00** | 0.47 | 0.64 | 0.83 | 0.65 | 0.73 |
| pbsv | 142X | 0.69 | 0.93 | 0.79 | **1.00** | 0.36 | 0.53 | 1.00 | 0.61 | **0.76** |
| Sniffles | 142X | 0.27 | **1.00** | 0.42 | 0.19 | **0.97** | 0.32 | 0.09 | 0.95 | 0.16 |
| SVIM | 142X | **0.97** | 0.94 | **0.96** | 0.97 | 0.50 | **0.66** | 0.15 | 0.65 | 0.25 |

The performance of each caller varied considerably by variant type and depth (Table 3). For deletions, SVIM had a considerably higher accuracy at 5X depth (F1-score = 0.85) compared to pbsv (F1-score = 0.53) and Sniffles (F1-score = 0.05). SVIM had the highest accuracy at 15X (F1-score = 0.95), followed by Sniffles (F1-score = 0.92), and pbsv (F1-score = 0.57). Sniffles had the highest accuracy at 30X depth (F1-score = 0.99), followed by SVIM (F1-score = 0.96) and pbsv (F1-score = 0.60). SVIM had the highest accuracy at 60X (F1-score = 0.96), followed by Sniffles (F1-score = 0.93) and pbsv (F1-score = 0.78). At 142X, SVIM had the highest accuracy (F1-score = 0.96), followed by pbsv (F1-score = 0.79) and Sniffles (F1-score = 0.42).

The accuracy of predicted duplications at 5X was higher for SVIM (F1-score = 0.45), compared to pbsv (F1-score = 0.36) and Sniffles (F1-score = 0.03). At 15X, 30X, and 60X depth, Sniffles had the highest accuracy (15X F1-score = 0.91; 30X F1-score = 0.98; 60X F1-score = 0.94) compared to pbsv (15X F1-score = 0.38; 30X F1-score = 0.46; 60X F1-score = 0.55) and SVIM (15X F1-score = 0.57; 30X F1-score = 0.61; 60X F1-score = 0.64). SVIM had the highest accuracy at 142X (F1-score = 0.66), followed by pbsv (F1-score = 0.53), and Sniffles (F1-score = 0.32). Both pbsv and SVIM had lower recall than precision, indicating that missed variant calls decreased the accuracy of these callers. Lower recall also decreased the accuracy of Sniffles at 5X, 15X, and 30X depth, while lower precision contributed more at 60X and 142X depth.

The accuracy of predicted inversions was higher in pbsv at 5X (F1-score = 0.69), 60X (F1-score = 0.75) and 142X depth (F1-score = 0.76). Sniffles had the highest accuracy at 15X (F1-score = 0.98) and 30X depth (F1-score = 0.92). Lower precision in SVIM at 5X (0.46) accounted for lower accuracy (F1-score = 0.49). At 15X, the SVIM precision and F1-scores increased to 0.84 and 0.74, respectively. The highest SVIM precision (0.92) and accuracy (F1-score = 0.76) was obtained at 30X. The SVIM precision decreased at 60X (0.83) and 142X depth (0.15), which resulted in lower accuracy (60X F1-score = 0.73; 142X F1-score = 0.25). At each depth, recall contributed more to decreased accuracy in pbsv. For Sniffles, lower recall contributed more to lower accuracy at 5X and 15X depth, while lower precision contributed more to lower accuracy at the higher depths.

## Agreement in predicted structural variants for wild *C. elegans* strains

To evaluate the usefulness of long-read DNA sequencing data to validate structural variants predicted from short-read technologies, we obtained data for 14 *C. elegans* wild strains with both Illumina and PacBio sequencing data.

The predicted variants varied considerably between the callers (Table 4). The predicted deletions ranged from a median of 76 per strain in SVIM to 341 in cnMOPs. Conversely, cnMOPs only predicted a median of 5.5 duplications per strain compared to 129 in Sniffles. The median predicted inversions per strain ranged from 8 in Lumpy to 88 in BreakDancer.

Because the exact breakpoints for the same predicted variant may differ between callers, it is difficult to directly compare the agreement of calls generated by different tools. Therefore, predicted variants spanning protein-coding genes were used to compare caller agreement. For each comparison between callers, only predictions of a given SV type that spanned the exact same set of genes were considered to be in agreement with each other. Overlapping predictions between the long-read callers and FusorSV were compared to evaluate using long-read sequencing data for the validation of variants predicted from short-read data.

Among the total set of predicted deletions spanning genes, 190 were shared among all long-read callers. Of these deletions, 119 were predicted by FusorSV and 64% of the genes spanned by deletions predicted by FusorSV were not shared by at least one long-read caller (Fig 4; S4 Table). Within the set of genes overlapping deletions predicted by SVIM, 95% were shared by at least one other caller. The other long-read tools had higher counts of unique predictions not found in any other caller (Assemblytics = 84%, MUM&Co = 33%, pbsv = 28%, sniffles = 45%, SVIM = 5%). The percentage of unique genes spanned by deletions also varied among the short-read callers (Fig 4; S5 Table) and ranged from 3% in DELLY to 80% in cnMOPS and CNVnator.

Among the total set of predicted tandem duplications spanning genes, only 66 were shared among all long-read callers. Of these only 52 were predicted by FusorSV (Fig 5; S6 Table). 77% of the genes spanned by duplications predicted by FusorSV were not shared by at least one long-read caller. Within the set of genes overlapping duplications, MUM&Co contained the

**Table 4. Predicted deletions, duplications, and inversions in *C. elegans*.**

| | Caller | Deletions | | Tandem Duplications | | Inversions | |
|---|---|---|---|---|---|---|---|
| | | Median deletions per strain | Median genes spanned by deletions per strain | Median duplications per strain | Median genes spanned by duplications per strain | Median inversions per strain | Median genes spanned by inversions per strain |
| Long-read tools | Assemblytics | 233.5 | 787 | 25 | 49.5 | N/A[1] | N/A[1] |
| | MUM&Co | 81 | 346 | 15.5 | 20 | 23.5 | 98.5 |
| | PBSV | 87 | 185 | 26 | 61.5 | 29 | 84 |
| | Sniffles | 171.5 | 385.5 | 129 | 393 | 63 | 206.5 |
| | SVIM | 76 | 119.5 | 71 | 112.5 | 21 | 61.5 |
| Short-read tools | FusorSV | 124.5 | 1363.5 | 128 | 1158 | 50 | 716.5 |
| | BreakDancer | 101 | 269.5 | 39.5 | 151.5 | 88 | 673 |
| | cnMOPs | 341 | 585.5 | 5.5 | 9.5 | N/A[1] | N/A[1] |
| | CNVnator | 269 | 2128.5 | 55 | 901 | N/A[1] | N/A[1] |
| | DELLY | 106 | 308 | 108 | 509.5 | 88 | 720 |
| | Hydra | 120.5 | 167.5 | 79.5 | 117 | 28.5 | 420 |
| | Lumpy | 107 | 340 | 107 | 551 | 8 | 36.5 |

1. Does not support inversions

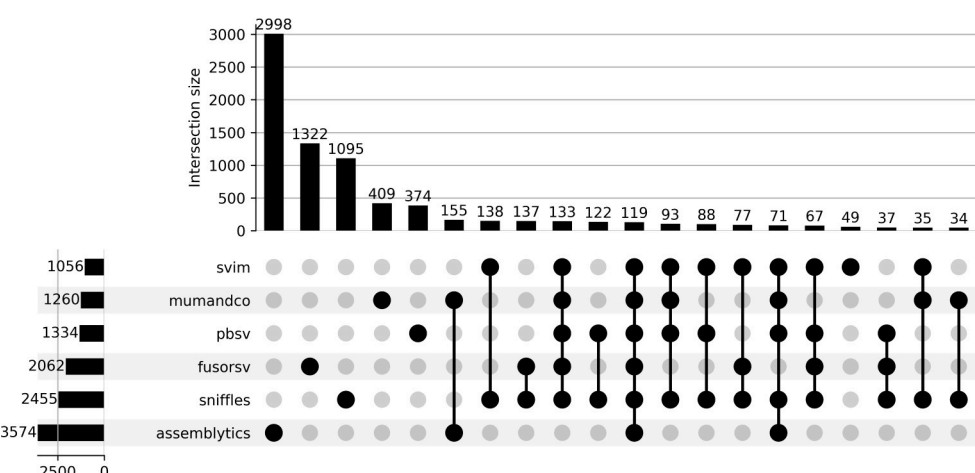

**Fig 4. Caller agreement for deletions spanning protein-coding genes predicted from long-read tools and FusorSV.**
Each row represents the set of genes covered by deletions for a given caller. The columns depict the intersection of
these predictions between callers. The plot is limited to the 20 largest sets.

fewest unique predictions (11%) followed by SVIM (14%), Assemblytics (20%), pbsv (42%),
and Sniffles (44%). The percentage of unique genes spanned by duplications also varied con-
siderably among the short-read callers (S7 Table) and ranged from 5% in DELLY to 90% in
CNVnator.

Among the total set of predicted inversions spanning genes, 59 were shared among all long-
read callers. Of these inversions, seven were predicted by FusorSV (Fig 6; S8 Table). Within
the set of genes overlapping inversions, SVIM contained the fewest unique predictions (41%)
followed by pbsv (51%), MUM&Co (60%), and Sniffles (69%). 89% of the genes spanned by
inversions predicted by FusorSV were not shared by at least one long-read caller. The percent-
age of unique genes spanned by inversions also varied considerably among the short-read call-
ers (S9 Table) and ranged from 17% in Lumpy to 47% in Hydra.

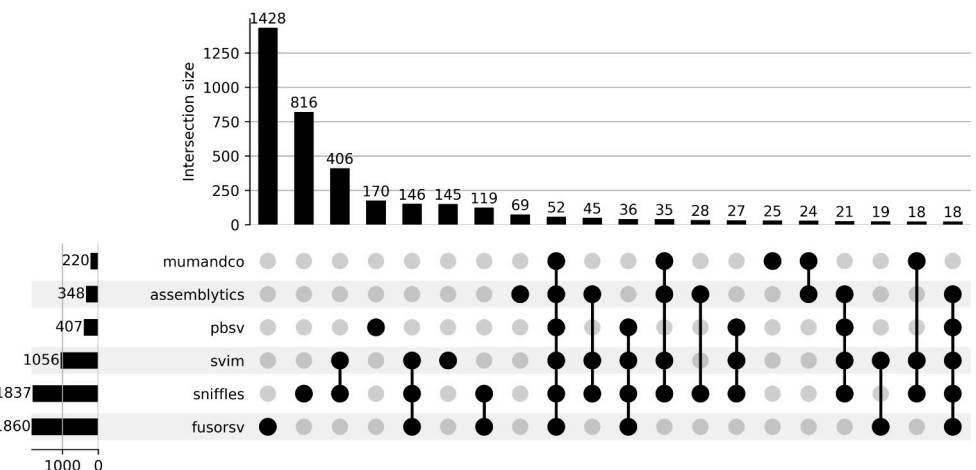

**Fig 5. Caller agreement for tandem duplications spanning protein-coding genes predicted from long-read tools
and FusorSV.** Each row represents the set of genes covered by duplications for a given caller. The columns depict the
intersection of these predictions between callers. The plot is limited to the 20 largest sets.

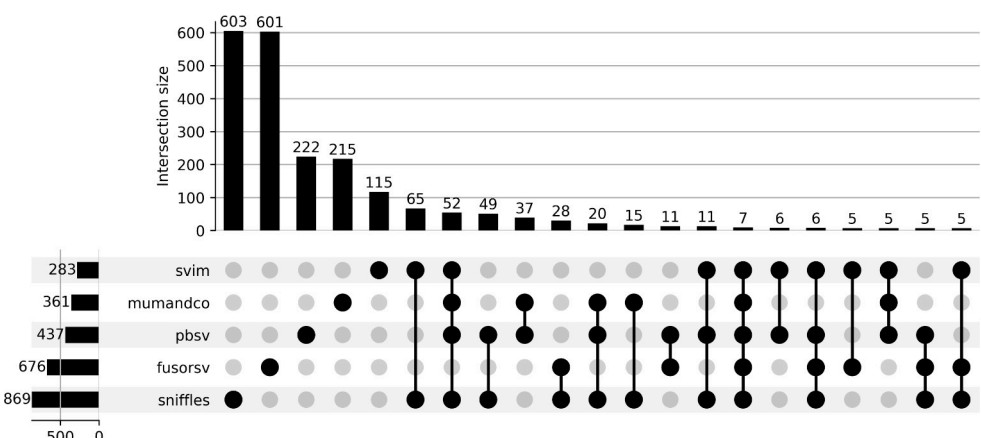

**Fig 6. Caller agreement for inversions spanning protein-coding genes predicted from long-read tools and FusorSV.** Each row represents the set of genes covered by inversions for a given caller. The columns depict the intersection of these predictions between callers. The plot is limited to the 20 largest sets.

## Discussion

Structural variant (SV) prediction is challenging for researchers studying non-human genomes. Most SV prediction tools were designed with the human genome in mind, and benchmarks on other species are lacking [13, 15, 16, 19]. Without an adequate guide for tool selection, the accuracy of predicted SVs has a high degree of uncertainty. To benchmark accuracy, we used simulated data to evaluate six short-read structural variant callers included in SVE [15], a pipeline developed to be used with FusorSV, an ensemble learning method that leverages the strengths of each individual caller. We further used real short- and long-read data to demonstrate the concordance or discordance between callers. We acknowledge that our selection of prediction tools is not comprehensive—a near impossible goal—and that other software has been released while we undertook this study.

The results for the simulated short-read data suggest that deletions and duplications may be predicted with high confidence using BreakDancer and DELLY, and accurate inversion predictions may be obtained using BreakDancer, DELLY, and Lumpy. The FusorSV performance typically reflected that of the best performing individual tools, but occasionally predicted large megabase spanning false positives. WGS from the 1000 Genomes Project (1000GP) [33] may be used to benchmark SV prediction using real data. These data provide a high-confidence human truth set, as SVs were validated using a combination of short- and long-read WGS data, Moleculo synthetic long-read sequencing, microarray SV detection, and targeted long-read sequencing. Our results are in discordance with the values reported in the literature for benchmarks generated using human data from the 1000 Genomes Project (1000GP) [15]. For example, the accuracy of deletion calls from 1000GP data were considerably lower for BreakDancer (F1-score = 0.47), DELLY (F1-score = 0.54), and FusorSV (F1-score = 0.62). Our results for duplications and inversions were substantially better than those reported for BreakDancer (duplication F1-score = 0.00, inversion F1-score = 0.08) DELLY (duplication F1-score = 0.01, inversion F1-score = 0.09), and FusorSV (duplication F1-score = 0.19, inversion F1-score = 0.45) based on 1000GP data. The Genome in a Bottle Consortium (GIAB) recently published a human SV benchmarking dataset containing 12,745 sequence resolved insertions (7,281) and deletions (5,464) [34]. These calls were generated from 19 variant calling methods using data from short-read, long-read, linked-read, optical, and electronic genome mapping. It should be noted that tandem duplications were categorized as insertions in these

data, which would lead to decreased performance in callers that discriminate between these two variant types. These data were used to benchmark Parliament2, a consensus based method that predicts SVs using support from multiple individual callers [16]. For the prediction of deletions from 35X depth short-read data, Parliament2 achieved an overall F1-score of 0.82, which included Delly (F1-score = 0.65), BreakDancer (F1-score = 0.59), Lumpy (F1-score = 0.50), and CNVnator (F1-score = 0.11) among the set of tools used for consensus calling. The performance of each of these tools was inferior to the performance we observed at 30X. Differences in genome properties, such as repeat content, could account in part for the improvements seen here, but the usage of simulated data was likely a major factor. Real data with experimentally validated variants would be valuable for identifying the underlying causes of these differences.

For the simulated PacBio data, the performance of each caller varied considerably by variant type and depth. For deletions, SVIM had the highest accuracy at 5X (F1-score = 0.85), 15X (F1-score = 0.95), 60X (F1-score = 0.96), and 142X depth (F1-score = 0.96), while Sniffles had the highest accuracy at 30X depth (F1-score = 0.99). For duplications, SVIM had the highest accuracy at 5X (F1-score = 0.45), and Sniffles had the higher accuracy at 15X (F1-score = 0.91), 30X (F1-score = 0.98), and 60X depth (F1-score = 0.94). Again, SVIM had the highest accuracy at 142X depth (F1-score = 0.66). For inversion calls, a higher accuracy was obtained using pbsv at 5X (F1-score = 0.69), 60X (F1-score = 0.75) and 142X depth (F1-score = 0.76). Sniffles had the highest accuracy at 15X (F1-score = 0.98) and 30X depth (F1-score = 0.92).

It should be noted that SVIM generates a VCF file containing all candidate SV calls, including calls of low confidence. Each prediction includes a quality score between 0 and 100 that provides a confidence estimate. The authors recommend using a threshold between 10–15 for higher depth datasets (*e.g.*, >40X) or a threshold that generates the expected number of predictions. Therefore, for datasets with lower sequencing depth, the analyst may be limited to selecting an arbitrary cut-off when the expected number of SVs is unknown. The thresholds we used for 5X (minimum QUAL = 2), 15X (minimum QUAL = 5), and 30X depth (minimum QUAL = 10) were proportional to the decrease in depth compared to the high depth specified by the SVIM authors. The performance for these cut-offs were similar to the optimum cut-off values that were calculated post-hoc (S10–S12 Tables), with the exception of inversions predicted at 142X depth, where a higher threshold is recommended.

Deletion benchmarks were previously described for pbsv and Sniffles using data from the Database of Genomic Variants [35] and NCBI dbVar [36] projects. The precision and recall were quantified for different numbers of reads supporting the deletions. Precision values up to 0.91 and 0.81 were reported for pbsv and Sniffles, respectively. Lower recall values were observed for pbsv (up to 0.45) and Sniffles (up to 0.26). By contrast, we observed lower precision but higher recall using simulated data.

Illumina and PacBio sequencing data from 14 natural *C. elegans* strains were analyzed to determine the concordance between predicted SVs among both short- and long-read callers. Low agreement was observed among all predictions generated using either Illumina or PacBio data. Furthermore, many SV calls unique to a single caller were observed for predictions made using either short or long-reads. It is therefore difficult to ascertain the accuracy of structural variants described in the literature, as the considerably different results may be generated using different tools. Nonetheless, the simulated data suggests that short-read tools, such as DELLY, BreakDancer, and Lumpy are likely to provide more accurate SV calling across a range of depths compared to the other short-read tools that were included in these benchmarks. If training data are available, FusorSV may also be used, but large megabase spanning inversions should be interpreted with caution, as several large false positives were predicted by this tool.

For SV prediction from long-reads, the simulated data suggested that neither pbsv, Sniffles, nor SVIM may be used with high-confidence for the prediction of duplications or inversions using lower depth data (e.g., 5X). Fewer generalizations can be made for higher depths. SVIM had the best performance for the prediction of deletions at 15X depth, while the performance of Sniffles was superior at 30X. Sniffles performed the best for the prediction of duplications and inversions at 15X and 30X depth. At 60X depth, SVIM had the highest accuracy for deletions, but Sniffles and pbsv demonstrated superior performance for duplications and inversions, respectively. At 142X depth, the SVIM accuracy was considerably higher than pbsv and Sniffles for deletions and duplications, while the accuracy of pbsv was considerably higher for inversions. If precision is less of a concern than recall, Sniffles may be the preferable choice for predicting duplications from long-read data.

The concordance between long-read callers is pertinent if long-read data is to be used to validate candidate SVs called using short-read data. Although few predicted SVs were common to all long-read callers, a large majority of the predicted deletions and duplications from SVIM were supported by at least one other caller. Therefore, SVIM might provide a more conservative option for validating deletions and duplications predicted from short-reads. Although inversions predicted by SVIM had the highest support among other callers, over one quarter of these calls were unique to SVIM. Therefore, long-read data may be less reliable for validating inversions predicted from short-reads.

To assess the concordance between short- and long-read approaches, the agreement between FusorSV and the long-read tools was measured. For each variant type, over three quarters of the FusorSV predictions were not shared by any of the long-read tools. Because higher accuracy has been reported for long-read tools in the literature, it is likely that FusorSV generated many false positives. Low agreement was observed for many of the SVs predicted by the individual tools used to train FusorSV despite higher accuracy observed in the simulated data. This may reflect a limitation in using simulated data to train FusorSV, as simulated data may bias the FusorSV models towards callers that perform poorly on real data. The low agreement observed between FusorSV and the long-read approaches is consistent with the results previously reported for CNV prediction in cattle [37]. In this study, the authors used CNVnator and Sniffles to predict deletions and duplications using Illumina and PacBio sequencing data. After filtering out probable false positives, only 18% of the CNVs predicted by CNVnator overlapped with those predicted by Sniffles. For CNVs spanning genes, 22% of the CNVnator calls overlapped with a Sniffles prediction.

The low agreement between different SV callers emphasizes the challenges involved in the selection of ideal tools for SV calling. Without high quality benchmarks in non-human species, tool choice is largely arbitrary, and the analyst will likely select a caller based on popularity. Despite the lack of benchmarks, researchers increasingly rely upon these tools for the characterization of SVs in non-human species, calling into question the reliability and reproducibility of past research findings. Improved benchmarks will provide an important resource for the research community.

## Conclusions

It is challenging to choose the appropriate tool for the prediction of structural variants from DNA sequencing data. Dozens of callers have been developed for calling structural variants using short-read data, but few independent benchmarks are available. Compounding this problem is the lack of benchmarks for non-human genomes. Here, multiple short-read and long-read callers were compared using both real and simulated *C. elegans* data. The results using simulated data showed that the performance of a given tool often varies considerably

according to variant type and sequencing depth and that no single tool performed best for all situations. The predictions generated from real data showed low overlap among all callers and many predictions unique to individual tools.

Because variants predicted from short-reads depend highly on the tool used, the analyst may choose to validate these SVs using long-read data. However, the lack of a consensus among long-read callers suggests that using a long-read caller to generate a "truth-set" warrants caution. Nonetheless, most of the deletions and duplications predicted by SVIM and MUM&Co, respectively, were shared by at least one other caller. These tools may provide a more conservative approach for validating SV calls using long-reads. The availability of reference datasets for which the "ground truth" is known would provide valuable resources for improving our understanding of the best approaches for SV prediction in non-human organisms. Future benchmarking projects would benefit from publicly available data from strains with precise deletions of various lengths generated using CRISPR-Cas9 methods, as well as further lab strains with SVs validated manually using long-read technologies and PCR.

## Methods

### Structural variation prediction

Structural Variation Engine (SVE) and FusorSV (v0.1.3-beta) [15] were used to predict structural variants (deletions, duplications, and insertions) from real and simulated short-read sequencing data. SVE is an SV calling pipeline that produces VCF files compatible with FusorSV. FusorSV uses an ensemble learning approach to call structural variants using a fusion model trained using individual callers. The six structural variant callers included in SVE that support non-human genomes were evaluated here: BreakDancer [38], cnMOPS [39], CNVnator [40], DELLY [41], Hydra [42], and Lumpy [22]. The default SVE and FusorSV parameter settings were used.

Five tools were used to predict structural variants (deletions, duplications, and insertions) from real and simulated long-read sequencing data: Assemblytics [43] (v1.2.1), MUM&Co [44] (v2.4.2), pbsv (https://github.com/PacificBiosciences/pbsv) (v2.6.2), Sniffles [45] (v1.0.12a), and SVIM [21] (v2.0.0). For each tool, the recommended long-read aligner and default parameter settings were used. The genome assemblies and alignments required for Assemblytics and MUM&Co were created in Canu [46] (v2.2) and MUMMER [47] (4.0.0rc1) respectively. The alignments used with pbsv were created using pbmm2 (v.1.7.0). The alignments used by Sniffles and SVIM were created using ngmlr [45] (v.0.2.7). Low confidence predictions below the SVIM quality score threshold were discarded using a different cut-off for each sequencing depth (5X = 2; 15X = 5; 30X = 10; 60X = 15, 142X = 15).

Custom Python (v3.7) and Bash scripts were used to select the final set of SV predictions to benchmark based on the following criteria: minimum size > = 100 bp, and vcf file FILTER flag = "PASS".

### Simulated data

SVsim (https://github.com/mfranberg/svsim; v. 0.1.1) was used to create mock *C. elegans* genomes containing simulated structural variants (deletions, duplications, and inversions) based on the WormBase [48] (Wbcel235; https://parasite.wormbase.org/Caenorhabditis_elegans_prjna13758/Info/Index/) reference assembly for the N2 strain. Because FusorSV uses the SV type and size as discriminating features to train the fusion model, the training dataset was designed to have 30 SVs of each type for each size/type bin. A total of 129 mock genomes with variable numbers of simulated deletions, duplications, and inversions were created ranging from 100bp to 280kbp. This ensured that the total base pairs spanned by SVs in each mock

genome did not exceed 1% of the reference genome, and that each bin was populated with a total of 30 SVs from the entire set of mock genomes. To benchmark the individual callers, a separate testing dataset, containing 129 mock genomes, was created with the same distribution of SV types and sizes.

Short-read DNA sequencing of the mock genomes was simulated with the randomreads.sh script included with BBTools (sourceforge.net/projects/bbmap; v38.79). Paired-end reads of 100bp were simulated at 5X, 15X, 30X, and 60X depths of coverage using the Illumina error model with default settings. SimLoRD [49] (v.1.0.4) was used to simulate PacBio sequencing data for the mock genomes. The SimLoRD PacBio sequencing runs were simulated at a depth of 5X, 15X, 30X, 60X, and 142X (the median depth of real PacBio data used to predict SVs using long-read tools).

### Real data

Data from the *Caenorhabditis elegans* Natural Diversity Resource (CeNDR) [23] were used to predict SVs in 14 *C. elegans* isolates collected from the wild. SV prediction using the SVE/FusorSV pipeline was performed using the BAM files provided for the 20200815 CeNDR release. SimuSCoP [50] (v1.0) was used to generate the simulated DNA-sequencing data that trained the FusorSV model. DNA sequencing of the *C. elegans* N2 reference strain were used to create the sequencing profiles used by SimuSCoP (SRA run = SRR3452263, SRA run = SRR1013928, SRA run = SRR9719854). For each strain, FusorSV models trained on simulated data of similar sequencing depth and read length were used to predict variants.

BC4586, a *C. elegans* strain containing validated structural variants, was used to evaluate the ability of the SVE/FusorSV pipeline in the prediction of experimentally validated structural variants. Simulated SimuSCoP DNA sequencing data was generated using an N2 profile (SRA run = SRR14489487) and used to train the FusorSV model. SVs for BC4586 (SRA run = SRR14489485) were predicted using the SVE/FusorSV pipeline and used to identify the presence of a deletion on chromosome IV (coordinates = 9853675–9857585), a tandem duplication on chromosome IV (coordinates = 9857585–9862397), and an inversion on chromosome IV (coordinates = 9853123–9853675).

### Structural variation benchmarking and comparison

Variant calling resulted in multiple overlapping structural variants of the same type, which can lead to inflated performance metrics, as each prediction may be counted as a true positive when compared to the truth dataset. For overlapping SV predictions of the same type and caller, a single call was selected using the criteria described in S13 Table. When no discriminating information was available among overlapping calls, the final SV was selected randomly.

**Benchmarking using simulated data.** For simulated data, each predicted variant was classified as being either a true positive (TP) or false positive (FP) using the Bedtools intersect command. Predictions that overlapped (minimum reciprocal overlap = 0.5) with at least one simulated variant in the mock genome were classified as true positives (TP), and those calls that did not were classified as false positives (FP). Simulated variants that were not predicted were classified as false negatives (FN). These classifications were used to calculate the following performance metrics:

Precision is the ratio of true positives (TP) to the total predicted variants, and was calculated as follows:

Precision = TP / (TP + FP)

Recall is the ratio of true positives to the total number of simulated variants, and was calculated as follows:

Recall = TP / (TP + FN)

The $F_1$ score provides a measure of the prediction accuracy by taking the weighted average of the precision and recall. The F1 score was calculated as follows:

$F_1$ = 2 * precision * recall/ (precision + recall)

Because the precision, recall, and F1 scores were calculated using binary classifications, they provide an incomplete picture. For each SV type, predictions meeting the minimum reciprocal overlap threshold are classified as true positives but may include calls with imprecise break-points or size estimates. Furthermore, no single agreed upon threshold exists for reciprocal overlap and the cutoff value is typically chosen arbitrarily. SVs may be called as multiple separate events that span the true variant [15], none of which meet the minimum reciprocal overlap requirement. If identifying genomic regions containing structural variation is of greater importance to the analyst compared to the precision of individual calls, metrics that are calculated using presence or absence classifications may be unsuitable.

For example, true positives are not penalized for predicting breakpoints outside of the region of the structural variant. Similarly, true positives are not penalized for predicting breakpoints within the true breakpoints of the structural variant. Therefore, the Jaccard index was calculated to measure the amount of overlap between the predicted variants and simulated variants.

The Jaccard index was calculated using the ratio of the number of base pairs in the intersection and union of the predicted and simulated variants:

Jaccard = (prediction variants ∩ simulated variants) / (prediction variants ∪ simulated variants)

Because the Jaccard score is calculated using the union and intersection of base pairs in the predicted and truth sets, it should be noted that this metric is biased towards the performance of larger variants.

## Comparisons using real data

Due to the lack of real data containing known structural variants, the sets of genes spanned by SVs predicted by each caller were compared to evaluate the consistency between different tools. *C. elegans* genome annotations obtained from WormBase [48] (release = WBPS14) were used to identify which genes were spanned by SVs in the CeNDR data. The gene set was limited to protein-coding genes where at least 50% of the gene was covered by an SV. SVs larger than 50 kbp were excluded due to this being the maximum size considered to be reliable in Assemblytics. For each comparison between callers, only predictions of a given SV type that spanned the exact same set of genes were considered to be in agreement with each other. The agreement between callers was depicted using UpSet plots [51].

## Supporting information

**S1 Table. Prediction of deletions from simulated short-read data.**
(XLS)

**S2 Table. Prediction of duplications from simulated short-read data.**
(XLS)

**S3 Table. Prediction of inversions from simulated short-read data.**
(XLS)

**S4 Table. Degree of overlap for deletions spanning genes predicted from long-read data and FusorSV.**
(XLSX)

**S5 Table. Degree of overlap for deletions spanning genes predicted from short-read data.**
(XLSX)

**S6 Table. Degree of overlap for duplications spanning genes predicted from long-read data and FusorSV.**
(XLSX)

**S7 Table. Degree of overlap for duplications spanning genes predicted from short-read data.**
(XLSX)

**S8 Table. Degree of overlap for inversions spanning genes predicted from long-read data and FusorSV.**
(XLSX)

**S9 Table. Degree of overlap for inversions spanning genes predicted from short-read data.**
(XLSX)

**S10 Table. Deletion performance for different SVIM QUAL cut-off thresholds.**
(XLS)

**S11 Table. Duplication performance for different SVIM QUAL cut-off thresholds.**
(XLS)

**S12 Table. Inversion performance for different SVIM QUAL cut-off thresholds.**
(XLS)

**S13 Table. Criteria used to select best overlapping call.**
(XLS)

## Acknowledgments

We thank Dr. Stephen Doyle and the other, anonymous reviewers for their comments.

## Author Contributions

**Conceptualization:** Kyle Lesack, James D. Wasmuth.

**Data curation:** Grace M. Mariene, Erik C. Andersen.

**Formal analysis:** Kyle Lesack.

**Funding acquisition:** James D. Wasmuth.

**Investigation:** Kyle Lesack.

**Methodology:** Kyle Lesack.

**Project administration:** James D. Wasmuth.

**Resources:** Erik C. Andersen.

**Software:** Kyle Lesack.

**Supervision:** Erik C. Andersen, James D. Wasmuth.

**Writing – original draft:** Kyle Lesack.

**Writing – review & editing:** Kyle Lesack, Grace M. Mariene, Erik C. Andersen, James D. Wasmuth.

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
