## [Decision Letter · Decision Letter 0]

6 Oct 2022

PONE-D-22-25734Different structural variant prediction tools yield considerably different results in Caenorhabditis elegansPLOS ONE

Dear Dr. Wasmuth,

Thank you for submitting your manuscript to PLOS ONE. After careful consideration, we feel that it has merit but does not fully meet PLOS ONE’s publication criteria as it currently stands. Therefore, we invite you to submit a revised version of the manuscript that addresses the points raised during the review process. Reviewer 1 raised several important concerns. We think they are essential issues that need to be addressed before considering the publication of the manuscript.  

We look forward to receiving your revised manuscript.

Kind regards,

Zechen Chong

Academic Editor

PLOS ONE

Journal Requirements:

"This work was supported by Results Driven Agricultural Research (RDAR #2016F013R) to JDW, the Natural Sciences and Engineering Research Council of Canada (NSERC) through Discovery Grants (#06239-2015 and 04589-2020) to JDW, and an NSERC Collaborative Research and Training Experience Program (CREATE) program in Host-Parasite Interactions (#413888-2012) to JDW and others. ECA was supported by a National Science Foundation CAREER award."

"DW; 04589-2020; The Natural Sciences and Engineering Research Council of Canada; https://www.nserc-crsng.gc.ca

JDW; 06239-2015; The Natural Sciences and Engineering Research Council of Canada; https://www.nserc-crsng.gc.ca

JDW; 413888-2012; The Natural Sciences and Engineering Research Council of Canada; https://www.nserc-crsng.gc.ca

JDW; 2016F013R; Results Driven Agricultural Research (Alberta); https://rdar.ca/

ECA; no number; National Science Foundation (USA); " ext-link-type="uri" xlink:type="simple">https://www.nsf.gov/"

4. Please clarify the Table 3 "Table 3 – Performance of long-read structural variant callers on simulated data" in page "9" and Table 3  "Table 3 – Predicted deletions, duplications, and inversions in C. elegans" in page "10".

5. Please upload a copy of Supporting Information Figure/Table/etc. Supplemental_Table_S1 to S_13 which you refer to in your text on pages 27 and 28.

Reviewers' comments:

Reviewer's Responses to Questions

**Comments to the Author**

1. Is the manuscript technically sound, and do the data support the conclusions?

Reviewer #1: Yes

Reviewer #2: Yes

Reviewer #3: Yes

2. Has the statistical analysis been performed appropriately and rigorously? 

Reviewer #1: N/A

Reviewer #2: Yes

Reviewer #3: Yes

3. Have the authors made all data underlying the findings in their manuscript fully available?

Reviewer #1: Yes

Reviewer #2: Yes

Reviewer #3: Yes

4. Is the manuscript presented in an intelligible fashion and written in standard English?

Reviewer #1: Yes

Reviewer #2: Yes

Reviewer #3: Yes

5. Review Comments to the Author

Reviewer #1: In this paper, authors benchmarked several short-read and long-read SV callers in simulated C. elegans datasets and tested these SV callers in several real datasets. In general, I do not think the conclusions of this paper would provide significant guidance for SV caller selection in future research. As no benchmark was available for real data, the authors showed low consistency between SV callers, from which we could not make any useful conclusions about which SV caller works best for C. elegans genome. Benchmarking SV callers only on simulated data is usually not sufficient, as some tools may perform very well in simulations but poorly in real data, depending on how the simulation (mock genome) is generated. Although I acknowledge the authors’ efforts in benchmarking all the SV callers at various sequencing depths, I would suggest rejecting current version of the manuscript and consider for publication if substantially revised.

Major concerns:

1. The interpretation of results in real data needs to be substantially revised. As no benchmark is available for real data, we could not use regular recall/precision/F1 score to assess SV discovery accuracy. Some alternative approaches should be used to compare the performance of SV callers, instead of simply showing low consistency between SV callers. There is much more work needed to be done after this point. The significance of this manuscript is largely determined by how SV callers are evaluated in real data without benchmark datasets. What the authors have presented are far from sufficient to answer the questions raised by authors at the beginning.

2. In the abstract, authors stated “multiple short-read and long-read tools were benchmarked using real and simulated data”. I would suggest revising such statements in the manuscript, as it is usually not considered as “benchmarked” in real data if no reference SV callset is available.

3. In the Introduction section, authors claimed that there are few public long-read datasets for non-human organisms. In fact, although benchmark datasets are relatively rare for non-human species, samples sequenced by both short-read and long-read platforms are pretty common in SRA. There are also projects like Vertebrate Genomes Project that really focuses on non-human organisms. I understand that C. elegans is the focus of this paper, but it may limit the audiences to C. elegans researchers, as benchmark results in C. elegans cannot represent other species if authors claim that benchmark in human cannot represent C. elegans.

4. In Table 3, it is quite surprising that long-read SV callers showed such low recall for duplications, except for Sniffles. Majority of the simulated SVs are shorter than 10kbp and should be reported by long reads. Duplications are sometimes considered as insertions by some SV callers. I know Sniffles, pbsv, and SVIM do report insertions in the VCF file. How were the insertion calls treated? If an insertion event is reported at a true duplication site with same SV size, should this duplication still be considered as FN?

5. In the section of ‘prediction of known structural variants’, only three previously validated SVs were used for benchmark short-read SV callers. Could the authors clarify if there are only three SVs in the benchmark dataset, or did they select just one SV from each SV type? Based on these three SVs in one sample, if they are not clinically relevant or extremely hard to identify, the results in this section are not significant enough to demonstrate which SV callers are more accurate, considering the fact that there are hundreds of SVs per C. elegans genome in last table.

6. The use of Jaccard index to represent SV calling accuracy could be a novel approach. According to its definition and its applications in simulation benchmarks, the Jaccard index could be biased to large SVs as they contain more numbers of base pairs. One thing we can try is to set a maximal allowed SV size when calculating Jaccard index. For example, if we set the cutoff at 2kbp, we only count 2000 bp for all SVs longer than the cutoff, so that a single 4Mbp FP will not reduce the Jaccard index a lot.

7. For the simulation benchmark in results section, I would suggest adding some more details about how mock genomes were generated, especially the number of SVs per genome. It would help readers who are not very familiar with C. elegans genome.

Minor concerns:

1. Why was Manta not included in the short-read SV caller comparison? It is also a widely used SV caller. In our previous experience of SV discovery in human genomes, Manta often performs better than other SV callers.

2. In the methods section, parentheses should be used in equations for Precision and Recall calculation. Precision = TP / (TP + FP) and Recall = TP / (TP+FN).

3. There are two ‘Table 3’ in the manuscript.

Reviewer #2: I appreciate the authors attention to the revisions and think this will be cited as a methodological caution to the field. I am glad to see the paper come out to the field. I have nothing more to add.

Reviewer #3: Lesack and colleagues describe an analysis of the ability of different bioinformatic tools to detect structural variation in simulated and real data from the model nematode Caenorhabditis elegans. The rationale is that there are several different tools and approaches to detect structural variants, but that existing benchmarks are based on human data which may not apply to other non-human species and that there is substantial variation in the precision and recall of different tools. The authors make use of C. elegans as it is one species for which there is a large amount of data (including that curated by one of the authors) available and some structural variants have been previously characterised; while this is not necessarily a “truth” dataset, it is perhaps as close as one can get for any non-human species.

The manuscript is well written and has been clearly improved and refined after having already undergone a round of peer review. Scientifically, it is my opinion that this study provides an important reflection on a decision-making process that is not well defined, even when a “gold standard” or “truth” dataset is used. The novelty that was perhaps underappreciated by the previous reviewers is that non-human organisms with different genome compositions and biases will behave quite differently from which the tools were designed and tested upon; this study is clear in pointing this out. While the study does not point to a clear winner, which is perfectly unsurprising, it does test several relevant parameters and does emphasise that more than one tool is needed. Perhaps a missing part of the discussion is the fact that some recent approaches for structural variant calling rely on the combination and consensus of multiple tools (eg. Parliment2, Zarate et al 2020 Gigascience, which is briefly mentioned in the introduction, but the overall concept is not revisited). Nonetheless, the point is not to develop yet another tool but to better understand how existing tools behave in different biological and experimental contexts. I think the authors have done a thorough job at this. In my opinion, the authors have also done a good job at addressing the previous reviewer's comments, made a number of sensible changes as well as pushed back on some unreasonable suggestions.

I am happy to support the publication of this manuscript as it is in PloS One.

Kind regards,

Stephen Doyle

Wellcome Sanger Institute

6. PLOS authors have the option to publish the peer review history of their article (what does this mean?). If published, this will include your full peer review and any attached files.

If you choose “no”, your identity will remain anonymous, but your review may still be made public.

Reviewer #1: No

Reviewer #2: No

Reviewer #3: **Yes: **Stephen Doyle

---

## [Author Response · Author response to Decision Letter 0]

8 Nov 2022

We thank each of the reviewers for the time that they spent reviewing the manuscript. We have sought to address their comments in the manuscript, providing clarification where necessary, both in the manuscript and our responses below.

**Editor

Acknowledgments Section: We have removed the funding information from the latest draft of the manuscript. The information given in the submission portal is correct.

Data Availability statement: We have updated the data availability statement in the submission portal. It is not feasible to make all the data available (it runs into terabytes). However, we have created a tutorial that allows any interested reader to recreate our simulated data. This is included on the GitHub repository.

Two table 3s: This is now fixed in the submitted manuscript.

SOM tables S1 to S13: These tables are now separated and uploaded.

**Reviewer 1

Reviewer’s comment 1: The interpretation of results in real data needs to be substantially revised. As no benchmark is available for real data, we could not use regular recall/precision/F1 score to assess SV discovery accuracy. Some alternative approaches should be used to compare the performance of SV callers, instead of simply showing low consistency between SV callers. There is much more work needed to be done after this point. The significance of this manuscript is largely determined by how SV callers are evaluated in real data without benchmark datasets. What the authors have presented are far from sufficient to answer the questions raised by authors at the beginning.

Authors’ response: On reflection, we agree that we incorrectly use the term benchmark when referring to real-world data and did not use recall/precision/F1 scores. We do not make claims as to the accuracy of the callers on real-world data. Our original plan had been to use the real PacBio data as a gold standard set for benchmarking. However, we were surprised to discover a high level of inconsistency between long-read SV callers. This led us to benchmark the long-read callers with simulated data. The only expert curated data for C. elegans is from Maroilley et al., which is used (also see our response to Comment 5 below). We have changed the manuscript text to make it clear where we are benchmarking with simulated data and where we are comparing the predictions of SV callers with real-world data. We had already discussed that benchmarking against real data is so far only possible with human-specific resources, such as the 1000 Genome Project.

Reviewer’s comment 2: In the abstract, authors stated “multiple short-read and long-read tools were benchmarked using real and simulated data”. I would suggest revising such statements in the manuscript, as it is usually not considered as “benchmarked” in real data if no reference SV callset is available.

Authors’ response: As in our response above, we agree with this comment and have changed the manuscript to distinguish between benchmarking and comparing predictions.

Reviewer’s comment 3: In the Introduction section, authors claimed that there are few public long-read datasets for non-human organisms. In fact, although benchmark datasets are relatively rare for non-human species, samples sequenced by both short-read and long-read platforms are pretty common in SRA. There are also projects like Vertebrate Genomes Project that really focuses on non-human organisms. I understand that C. elegans is the focus of this paper, but it may limit the audiences to C. elegans researchers, as benchmark results in C. elegans cannot represent other species if authors claim that benchmark in human cannot represent C. elegans.

Authors’ response: Here, we presume that the reviewer is referring to the following sentence: At time of writing, most curated, large, multi-population, whole genome datasets for metazoan species are restricted to humans. We apologise if this statement is confusing. The key phrase here is ‘multi-population’. For this work, we need several distinct isolates of a given species that have both short- and long-read data, with the associated metadata. At the time we started this work, we found only C. elegans fit our criteria; we had looked at the VGP. Given the reviewer’s comments, we revisited the VGP. We went through the SRA data for the VGP umbrella project (https://www.ncbi.nlm.nih.gov/bioproject/489243) and identified 762 genomic sequencing runs from 33 species (there were other species with just transcriptome data under this project). From all the metadata we could not find a species where multiple, clearly defined isolates had been sequenced. We anticipate the flood of these population WGS data in the next few years. In addition to the VGP, we looked at cattle sequencing data and discovered that some of our concerns have been raised in cattle (https://www.sciencedirect.com/science/article/pii/S0022030217303521). The authors show little overlap between short- and long-read SV predictions, though do not explore these trends as deeply as we do. We agree that we can only present a C. elegans-centric view. However, we believe that the results of our study, given the robustness of the C. elegans data, will encourage people to use SV callers with caution whether using invertebrate or non-human vertebrates in the future.

Reviewer’s comment 4: In Table 3, it is quite surprising that long-read SV callers showed such low recall for duplications, except for Sniffles. Majority of the simulated SVs are shorter than 10kbp and should be reported by long reads. Duplications are sometimes considered as insertions by some SV callers. I know Sniffles, pbsv, and SVIM do report insertions in the VCF file. How were the insertion calls treated? If an insertion event is reported at a true duplication site with same SV size, should this duplication still be considered as FN?

Authors’ response: We were also surprised. Further, it is an understandable assumption that “Majority of the simulated SVs are shorter than 10kbp and should be reported by long reads." However, it is not what we see when using the long-read SV callers. The reviewer makes an excellent observation that “[d]uplications are sometimes considered as insertions.” We do not seek to treat them as special cases for a couple of reasons. First, our motivation here is to show what a typical researcher would see when running these software. Second, each of the longread callers is capable that we evaluated is capable of discriminating between novel sequence insertions and tandem duplications. If interspersed duplications are present, the duplicated sequence could in theory be categorized as insertions by callers whose duplication calls are limited to those that occur in tandem. Of the longread callers that were evaluated, only SVIM is able to predict interspersed duplications. Duplications misclassified as insertions could be potentially be identified by mapping the inserted sequences against the reference genome, however, this goes beyond what the typical analyses that a typical researcher would do when using this software. We also excluded any interspersed duplications predicted by SVIM, as this SV type was not supported by any other caller.

We have revised the manuscript to describe our reasons for excluding insertions and to clarify that the duplication category only includes tandem duplications. 

Reviewers’ comment 5: In the section of ‘prediction of known structural variants’, only three previously validated SVs were used for benchmark short-read SV callers. Could the authors clarify if there are only three SVs in the benchmark dataset, or did they select just one SV from each SV type? Based on these three SVs in one sample, if they are not clinically relevant or extremely hard to identify, the results in this section are not significant enough to demonstrate which SV callers are more accurate, considering the fact that there are hundreds of SVs per C. elegans genome in last table.

Authors’ response: In response to the recommendation of a previous reviewer, we sought a dataset of validated SVs. To the best of our knowledge, Maroilley et al. is the only in C. elegans where predicted SVs have undergone expert review. From this study, we removed short SVs (50bp) or complex SVs (inverted tandem duplications). We accept that it is a limited survey, but we consider it informative for the reader.

Reviewer’s comment 6: The use of Jaccard index to represent SV calling accuracy could be a novel approach. According to its definition and its applications in simulation benchmarks, the Jaccard index could be biased to large SVs as they contain more numbers of base pairs. One thing we can try is to set a maximal allowed SV size when calculating Jaccard index. For example, if we set the cutoff at 2kbp, we only count 2000 bp for all SVs longer than the cutoff, so that a single 4Mbp FP will not reduce the Jaccard index a lot.

Author’s response: Our use of the Jaccard index was a focus on recommendations by an earlier reviewer and we made changes to the manuscript submitted to PLOS One. We have previously considered setting a maximal allowed SV size, but any cut-off will always be arbitrary. However, it response to the above comment, we have elaborated on the strengths and weaknesses of using the Jaccard index in the methods, results, and discussion sections.

Reviewer’s comment 7: For the simulation benchmark in results section, I would suggest adding some more details about how mock genomes were generated, especially the number of SVs per genome. It would help readers who are not very familiar with C. elegans genome.

Authors’ response: We agree with the recommendation and have included more information in the manuscript.

Reviewer’s comment 8 (minor): Why was Manta not included in the short-read SV caller comparison? It is also a widely used SV caller. In our previous experience of SV discovery in human genomes, Manta often performs better than other SV callers.

Authors’ response: As with any benchmarking/comparison paper, it is almost impossible to capture all methods. One paper describes 69 different tools. We selected short-read SV predictors that were readily compatible with SVE/FusorSV pipeline, which does not yet include Manta. We also considered the most cited methods: DELLY (1556 citations), BreakDancer (1511 citations), CNVnator (1409 citations), and LUMPY (1074 citations). By comparison, Manta has 1031 citations at present (October 21st, 2022). While on par with LUMPY, feasibility meant that we had to draw the line somewhere. In the discussion, we now state explicitly that our comparison is not exhaustive. 

Reviewer’s comment 9 (minor): In the methods section, parentheses should be used in equations for Precision and Recall calculation. Precision = TP / (TP + FP) and Recall = TP / (TP+FN).

Authors’ response: Thank you for pointing out this mistake, it has been fixed.

Reviewer’s comment 10 (minor): There are two ‘Table 3’ in the manuscript.

Authors’ response: Thank you for pointing out this mistake, it has been fixed.

**Reviewer 2

Reviewer’s comment: I appreciate the authors attention to the revisions and think this will be cited as a methodological caution to the field. I am glad to see the paper come out to the field. I have nothing more to add.

Authors’ response: We thank the reviewer for taking the time to review the manuscript for a second time and for considering our revisions to previous reviews. 

**Reviewer 3

Reviewer’s comment (we assume minor): Perhaps a missing part of the discussion is the fact that some recent approaches for structural variant calling rely on the combination and consensus of multiple tools (eg. Parliment2, Zarate et al 2020 Gigascience, which is briefly mentioned in the introduction, but the overall concept is not revisited).

Authors’ response: We thank Dr. Doyle for all his positive comments on the manuscript. With regard to the one above, we agree that a fuller discussion on combination approaches. We have elaborated on this in an improved discussion.

---

## [Decision Letter · Decision Letter 1]

16 Nov 2022

Different structural variant prediction tools yield considerably different results in Caenorhabditis elegans

PONE-D-22-25734R1

Dear Dr. Wasmuth,

We’re pleased to inform you that your manuscript has been judged scientifically suitable for publication and will be formally accepted for publication once it meets all outstanding technical requirements.

Kind regards,

Zechen Chong

Academic Editor

PLOS ONE

Additional Editor Comments (optional):

Reviewers' comments:

Reviewer's Responses to Questions

**Comments to the Author**

1. If the authors have adequately addressed your comments raised in a previous round of review and you feel that this manuscript is now acceptable for publication, you may indicate that here to bypass the “Comments to the Author” section, enter your conflict of interest statement in the “Confidential to Editor” section, and submit your "Accept" recommendation.

Reviewer #1: All comments have been addressed

Reviewer #2: All comments have been addressed

Reviewer #3: All comments have been addressed

2. Is the manuscript technically sound, and do the data support the conclusions?

Reviewer #1: Yes

Reviewer #2: Yes

Reviewer #3: (No Response)

3. Has the statistical analysis been performed appropriately and rigorously? 

Reviewer #1: (No Response)

Reviewer #2: Yes

Reviewer #3: (No Response)

4. Have the authors made all data underlying the findings in their manuscript fully available?

Reviewer #1: Yes

Reviewer #2: Yes

Reviewer #3: (No Response)

5. Is the manuscript presented in an intelligible fashion and written in standard English?

Reviewer #1: Yes

Reviewer #2: Yes

Reviewer #3: (No Response)

6. Review Comments to the Author

Reviewer #1: (No Response)

Reviewer #2: I was happy with the previous revisions and thought the paper could have been accepted in the previous round.

I do want to comment on some of the requests from the other reviewer and to validate some of the authors' responses. First, PLoS One is supposed to accept papers that are technically sound, regardless of impact. I support the authors' choice not to expand manuscript with analyses outside the scope of their work.

Second, in our own work on SVs we have found discordance in SV callers and have found that those tuned to human variation perform poorly in model organisms. We have less experience in C. elelgans but can state that we find similar results using these same methods in Drosophila also with moderate sized CNVs. pbsv performed especially badly (10% identified) and the documentation was not sufficient for us to discern why. The results here echo our own frustration that greater method development is needed-- and possibly organism specific bioinformatic pipelines. It is nice to see a group run through the pitfalls, and while the manuscript makes modest advances in solving the problem, this paper will offer information that is useful in the field.

Reviewer #3: (No Response)

7. PLOS authors have the option to publish the peer review history of their article (what does this mean?). If published, this will include your full peer review and any attached files.

Reviewer #1: No

Reviewer #2: No

Reviewer #3: **Yes: **Stephen R Doyle

---

## [Editor Report · Acceptance letter]

2 Dec 2022

PONE-D-22-25734R1 

Different structural variant prediction tools yield considerably different results in *Caenorhabditis elegans*

Dear Dr. Wasmuth:

I'm pleased to inform you that your manuscript has been deemed suitable for publication in PLOS ONE. Congratulations! Your manuscript is now with our production department. 

Kind regards, 

on behalf of

Dr. Zechen Chong 

Academic Editor

PLOS ONE